# Sleep problems and chronic conditions in single parents in Ghana: Serial mediating roles of health-related quality of life and functional limitations

Obed Jones Owusu-Sarpong[1]*, Kabila Abass[1], Daniel Buor[1], Solomon Osei Tutu[2], Razak M. Gyasi[3,4]

1 Department of Geography and Rural Development, Kwame Nkrumah University of Science and Technology, Kumasi, Ghana, 2 Department of Social Science, Offinso College of Education, Offinso, Ashanti Region, Ghana, 3 African Population and Health Research Center, Nairobi, Kenya, 4 National Centre for Naturopathic Medicine, Faculty of Health, Southern Cross University, Lismore, NSW, Australia

* obedjones339@gmail.com

**Data Availability Statement:** Data have been uploaded as a supplementary file. For more information please contact the corresponding

## Abstract

### Objectives

Data on the association between sleep problems and chronic conditions among single parents in low- and middle-income countries (LMICs) are limited, and no study has, to date, reported the serial mediation effects of functional limitations and poor health-related quality of life (HRQoL) in this association. This study examines the extent to which functional limitations and poor HRQoL serially explain the link between sleep problems and chronic conditions among single parents in Ghana.

### Methods

Data on 627 single mothers and fathers were obtained through a multi-stage stratified sampling technique. Sleep duration, nocturnal sleep problems, and daytime sleep problems were used to assess sleep problems. The EQ-5D-3L questionnaire was used to measure HRQoL. Multivariable OLS models and bootstrapping serial mediation analyses were performed to evaluate the hypothesized associations.

### Results

The mean age (SD) was 45.0 (14.66) years; 67.3% females. After full adjustment, sleep problems were significantly associated with increases in chronic conditions (β = .238, 95% CI = .100-.377), poor HRQoL (β = .604, 95%CI = .450-.757), and functional limitations (β = .234, 95%CI = .159-.307). Chronic conditions were positively influenced by poor HRQoL (β = .352, 95%CI = .284-.421) and functional limitations (β = .272, 95%CI = .112-.433). Sleep problems were indirectly related to chronic conditions via poor HRQoL (β = .213, Boot*SE* = .039, 95%CI = .143-.295), functional limitations (β = .063, Boot*SE* = .029, 95%CI = .013-

author Tel: +233245849372 Email:
obedjones339@gmail.com.

**Funding:** The author(s) received no specific
funding for this work.

**Competing interests:** authors have no competing
interests.

.130) and functional limitations → HRQoL (β = .099, Boot*SE* = .025, 95%CI = .054-.152),
mediating 34.70%, 10.31% and 16.15% of the total effect, respectively.

## Conclusions

Sleep problems and poor HRQoL were positively associated with chronic conditions. Functional limitations and poor HRQoL partially and serially explained this association. Efforts to address chronic conditions among single parents should consider interventions for sleep problems and physiological health outcomes, particularly in LMICs.

## Introduction

Sleep problems (encompassing difficulties in both sleep quality and quantity), such as insomnia, sleep apnea, and irregular sleep patterns, are increasingly prevalent worldwide [1]. Sleep problems have been linked to a myriad of adverse health outcomes, including the development and exacerbation of chronic conditions such as cardiovascular disease, diabetes, obesity, and mental health disorders [2–4]. In recent years, the relationship between sleep health and physical well-being has garnered increasing attention within the field of public health and medicine. Adequate and quality sleep is essential for various bodily functions, including cognitive performance, immune system regulation, and metabolic processes [5–7]. Sleep deprivation and poor sleep quality, however, disrupt hormonal regulation, immune function, and metabolic processes, predisposing individuals to a range of chronic diseases [2, 3].

Single parents, a vulnerable population group, often face unique challenges that can contribute to the development of sleep problems and chronic health issues. There is a growing prevalence of single-parent households globally and Ghana in particular [8], and therefore, understanding the relationship between sleep problems and chronic conditions among this population is of paramount importance [9]. Single parents may face increased stress in caregiving responsibilities, financial constraints, social isolation, and limited access to healthcare, all of which can negatively impact their sleep and overall health [10]. However, the intersection of sleep health and chronic conditions represents a critical yet understudied area within the realm of public health, particularly in the context of vulnerable populations such as single parenthood in low- and middle-income countries (LMICs), thereby hindering our understanding of this complex interplay, especially within diverse cultural contexts.

Recent research has shed light on the complex interplay between sleep problems and chronic conditions. For instance, Gao et al. [11] assert that short sleep duration is causally linked to an increased risk of rheumatoid arthritis (RA). In a more detailed explanation, Medic et al. [6] showed that sleep disruption is associated with increased activity of the sympathetic nervous system and hypothalamic–pituitary–adrenal axis, metabolic effects, changes in circadian rhythms, and pro-inflammatory responses. Indeed, long-term consequences of sleep disruption in otherwise healthy individuals include hypertension, dyslipidemia, cardiovascular disease, weight-related issues, metabolic syndrome, type 2 diabetes mellitus, and colorectal cancer [6].

Most previous studies of sleep problems and their health consequences have focused on individual diseases such as hypertension, type 2 diabetes, or depression [11–14]. This current study, however, explored the association between sleep problems and the risk of 11 different types of chronic conditions among single parents in an LMIC like Ghana, where data on such vulnerable populations is scarce. The relationship between sleep problems and chronic

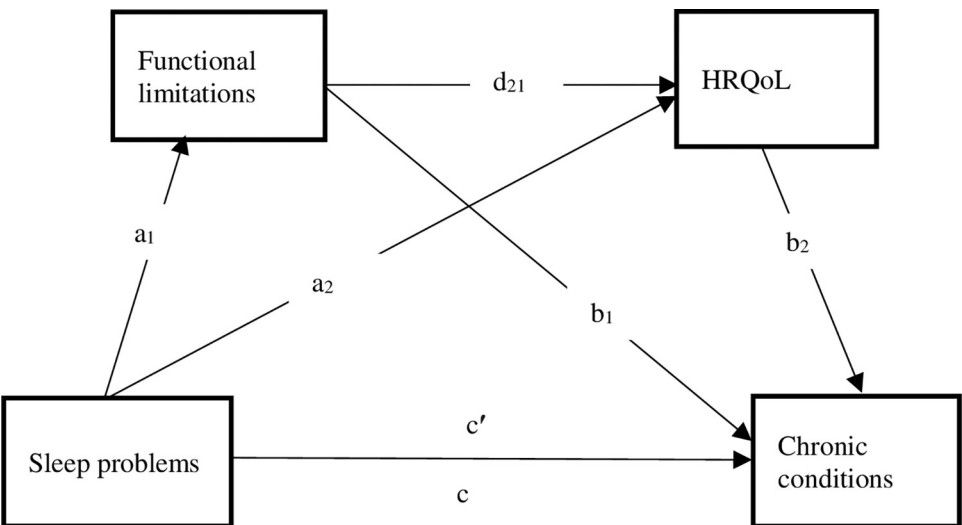

**Fig 1. A hypothesized partial mediation model of the effect of sleep problems on chronic conditions via HRQoL and functional limitations as potential mediating variables.** Paths $a_1$, $a_2$, $b_1$, $b_2$, c, and c′, denote path coefficients. Hypothesis 1: Sleep problems positively affect chronic conditions (total effect, c). Hypothesis 2: There is a significant specific indirect effect of Sleep problems on chronic conditions via HRQoL (indirect effect, $a_1b_1$). Hypothesis 3: Sleep problems have a significant, specific indirect effect on chronic conditions via functional limitations (indirect effect $a_2b_2$). Hypothesis 4: Sleep problems have a significant cascading or serial mediation effect on chronic conditions via functional limitations and HRQoL (serial indirect effect $a_1d_{21}b_2$).

conditions among single parents in Ghana is complex and may be mediated by other factors such as health-related quality of life (HRQoL) [15, 16] and functional limitations [17, 18]. Empirical evidence suggests that sleep problems may lead to functional limitations [19, 20], and functional limitations may lead to chronic conditions [21, 22]. Again, sleep problems may lead to poor HRQoL [23, 24], and poor HRQoL can also lead to chronic conditions [25, 26]. This study further examined the serial mediation role of functional limitations and poor HRQoL in the association of sleep problems and chronic conditions.

By examining the complex interplay between sleep problems, chronic conditions, functional limitations, and HRQoL, this research has the potential to contribute to the growing body of knowledge on the health and well-being of single parents in Ghana and other similar contexts. See Fig 1 for the study's hypothesized serial multiple mediation model of the effect of sleep problems on chronic conditions via functional limitations and HRQoL as potential mediating variables. We hypothesized that there would be significant positive association between sleep problems and chronic conditions. We further hypothesized that the positive association between sleep problems and chronic conditions would be serially mediated by functional limitations and poor health-related quality of life.

## Methods

### The survey

Multi-stage stratified sampling was used in this study. Atwima Kwanwoma District was purposively selected based on anecdotal evidence, which suggests that the population of single parents is high in the district. The district was unofficially divided into Atwima and Kwanwoma, and the various communities were stratified into urban and rural based on their socioeconomic features and general development levels like economic activity levels, nature of roads, health, and educational infrastructure [8]. A simple random sampling technique was

employed to select 12 communities, 4 urban and 8 rural, out of the total of 64 communities in the district [27]. Names of all the respective communities were written on paper, taking into cognizance the fact that some communities are in Atwima and others are in Kwanwoma. This is an unofficial division of the district. Naturally, the district is divided into East and West.

By the fishbowl selection method, a total of 6 communities (4 rural and 2 urban) were chosen from Atwima, and another 6 communities (4 rural and 2 urban) were chosen from Kwanwoma, for a total of 6 communities, resulting in a grand total of 12 communities (4 urban and 8 rural). The basis for the selection of more rural communities (8 communities) than urban communities (4 communities) is due to the fact that the urban communities are more populated than the rural communities. A default or consecutive prevalence of 50% [28], which translates to 0.5, was used as a sampling frame for determining the number of single parents in the district. This is because no information is available on the number of single parents in the district. Based on the sampling frame, Fisher's sample size estimation formula was employed to determine the overall sample size for the study [29].

In this study, the minimum sample size was achieved using the following estimation parameters: $\propto$ (95% CI) = 1.96, the consecutive or default prevalence, $p$ = 50% = 0.5 (and given that the proportion of single parents aged $\geq$13 years in the district is currently unknown), $d$ = 5% = 0.05, and 5% type 1 error and 15% type 2 error. We then adjusted the minimum sample size for anticipated sample defects such as non-responses and outliers and to improve the generalizability of findings [30]. This was also crucial to allow rigorous analytic procedures. Therefore, the final analytic sample in this study is $N$ = 627. The researchers developed a face-to-face interviewer-administered questionnaire to reach out to the respondents. To help administer the questionnaire, five research assistants were trained by the researchers for a period of five days, starting from Monday, 6th June to Friday, 10th June 2023. The participants' recruitment period for this study started from 20th June 2023 to 19th August 2023.

### Ethical consideration

The study was carried out in accordance with the World Medical Association's Declaration of Helsinki's code of ethics for research involving human subjects. As a result, the study protocol was reviewed and approved by the Humanities and Social Sciences Research Ethics Committee (HuSSREC) at Kwame Nkrumah University of Science and Technology in Kumasi, Ghana (Ref number: HuSSREC/AP/111/VOL.1.) Additionally, written informed consent was obtained from each respondent. Indeed, we sought permission and approval from the parents or guardians of a few minors who had less than 18 years during data collection. Moreover, in the Ghanaian context, you are considered an adult when you give birth and assume taking care of others or your children.

### Measures

### Independent variable

**Sleep problems.**   To assess sleep disorders, three sub-domains were used: sleep duration, nocturnal sleep troubles, and daytime sleep problems. In response to the inquiry, "On average, how many hours of sleep do you get in a 24-hour period over the past month?" sleep length was recorded in hours of sleep. A continuous scale was employed to record the answers. The study employed a fatigue-related questionnaire to measure sleep problems at night. The findings indicate a moderate-to-high sensitivity for detecting cases of obstructive sleep apnea in the general population that are clinically significant [31]. The respondents were asked, "Overall, in the last 30 days, how much of an issue did you have with sleeping, e.g., falling asleep, waking up during the night, or waking up too early in the morning?" Daytime sleep concerns

were measured using the following question: "Overall, in the last 30 days, how much of a problem did you have due to not feeling rested and refreshed during the day (for example, feeling tired, not having energy)?" Five response options were available for these questions: none, mild, moderate, severe, and extreme (coded from 1 to 5). These items have been used in prior sub-Saharan Africa (SSA) data investigations [32]. An increasing score indicated more issues with both daytime and nighttime sleep, leading to the generation of a latent poor sleep quality (PSQ) score [13, 33]. The variable comprising just the first two questions achieved a Cronbach's Alpha of 0.532.

## Dependent variable

**Chronic conditions.** Our dependent variable was chronic conditions, which consist of 11 chronic physical conditions. Participants were asked whether a healthcare professional had ever told them they had any chronic physical conditions. A list of the 11chronic physical conditions, including hypertension, diabetes, heart disease, stroke, cancer, renal disease, chronic respiratory diseases, asthma, arthritis/rheumatism, and ulcers, was presented to the participant, and the count of chronic diseases, included the presence (1) or absence (0). This was used to evaluate chronic physical conditions.

## Mediators

**Health-related quality of life–HRQOL.** The EQ-5D-3L questionnaire was used to measure the HRQoL variable. It consisted of five items related to mobility, self-care, daily activities, pain/discomfort, and restlessness. A three-level scale was used to code the replies for each indicator, with 1 denoting no problems, 2 moderate problems, and 3 severe problems. This ranged from 5 to 15, where increasing levels suggested poor HRQoL.

**Functional status.** To determine functional limitations, single parents were asked to self-report how difficult it was for them to perform activities of daily living (ADL) and instrumental activities of daily living (IADL). A popular metric for assessing an individual's daily performance is difficulty performing IADLs and ADLs (WHO, 2012). Six-item questions representing daily performance on two levels—no = 1 or yes = 2—were used to evaluate ADL. Higher total scores indicated higher degrees of ADL; the range of scores was 6 to 12. Additionally, five criteria pertinent to the local situation were used to measure IADL. Higher scores denoted higher degrees of IADL. The total score varied from 5 to 10. The scores of these subdomains were merged and standardized to create the functional limitations variable. Cronbach's Alpha for the variable functional restriction was 0.901.

**Covariates.** The analytic models were adjusted with demographic, socio-economic, lifestyle, and health-related variables selected from previously published literature [34, 35]. Sex (male or female), residential status (rural/urban), educational attainment (none, primary, secondary, tertiary), work status (unemployed/employed), and income levels (in Ghana Cedis), religion (religious or non-religious) are among the socio-economic and demographic characteristics. As a control variable, physical activity participation, as determined by the International Physical Activity Questionnaire, was added. Additional inclusions were current smoking (no = 0/yes = 1) and alcohol intake (no = 0/yes = 1), National Health Insurance Scheme (NHIS status) (yes or no). S1 Dataset provides the dataset associated to these stusy variables.

## Statistical analyses

The statistical analysis was conducted using Statistical Package for Social Scientists (SPSS) v25 (SPSS, Inc., IBM, Armonk, NY, USA). First, descriptive statistics were calculated to describe

the sample. We estimated means and standard deviation for the continuous variables and counts and percentages for the categorical variables. Second, zero-order correlations (Pearson's r) were performed between sleep problems, chronic conditions, HRQoL, and functional limitations to evaluate correlations and the bivariate associations between these key variables.

We evaluated the hypothesized serial mediation path models using the PROCESS macro plug-in [36], where sleep problems were the independent variable, HRQoL and functional limitations were the mediators, and chronic conditions were the dependent variable. We conducted a non-parametric bootstrapping procedure to compute confidence intervals (CIs) around the indirect effect. The final estimate of the indirect effect is represented by the mean indirect effect computed across 5,000 bootstrap samples. This estimation method involves random and repeated sub-sampling, with greater statistical power than a traditional Baron and Kenny's casual mediation estimation approach [37]. The models were adjusted for sex, physical activity, smoking status, residential type, alcohol intake, depression, NHIS status, income, religion, and education status. The statistical significance of the indirect effects was examined based on bias-corrected confidence intervals derived from 5000 bootstrap samples [36]. The level of statistical significance was set at $P$ value < 0.05 for the indirect effect if its 95% bootstrap CI does not overlap zero.

## Results

### Sample characteristics

A total of 627 individuals were included in the current study. The characteristics of this analytic sample are shown in Table 1. The average age was 45.0 (SD = 14.66) years, ranging from 14 to 85 years, and 67.3% were females. The majority of the sample lived in rural communities (54.5%), were employed (71.8%), and neither smoked (97.4%) or consumed alcohol (92.5%). On average, every participant had schooled for 9.5 (SD = 4.0) years while the mean monthly income was approximately 1938.28 Ghana Cedis (USD$ 140) (SD = 904.46). Moreover, the mean physical activity, social network, and number of chronic physical conditions scores were 330.79 MET min (SD = 420.83), 8.32 (SD = 6.44), and 1.07 (SD = 1.35), respectively.

### Correlations

Zero-order Pearson's correlation matrix is presented in Table 2. The results indicated that the sleep problem score was significantly related to all key independent and potential mediating variables. Thus, sleep problem was positively correlated with chronic conditions ($r = -.371$, $p < .001$), functional limitations ($r = -.273$, $p < .001$), and poor HRQoL ($r = -.436$, $p < .001$). Also, chronic conditions positively correlated with both poor health-related quality of life ($r = .565$, $p < .001$) and functional limitations ($r = .457$, $p < .001$), while poor HRQoL correlated positively with functional limitations ($r = .576$, $p < .001$). The significant correlations between sleep problems, chronic conditions, poor HRQoL, and functional limitations provide a firm ground for statistical assumptions for further serial mediation modeling of the indirect effect of sleep problems and chronic conditions in this sample.

### Serial mediation analysis

Fig 2 presents the unstandardized regression coefficients in each hypothesized pathway. Indeed, all hypothesized path models were statistically significant even after a robust adjustment for several potential covariates. The results found that sleep problems were significantly associated with increases in chronic conditions (β = .238, 95%CI = .100-.377), poor HRQoL (β = .604, 95%CI = .450-.757), as well as functional limitations (β = .234, 95%CI = .159-.307).

**Table 1. Sample characteristics (*N* = 627).**

| | Variable | Valid N | (%) | Mean | (*SD*) |
|---|---|---|---|---|---|
| Age (in years) (range: 14 to 85) | | | | 44.95 | (14.66) |
| Sex (n, %) | | | | | |
| | Male | 205 | (32.7) | | |
| | Female | 422 | (67.3) | | |
| Residential type (n, %) | | | | | |
| | Rural | 342 | (54.5) | | |
| | Urban | 285 | (45.5) | | |
| Employment status (n, %) | | | | | |
| | Unemployed | 137 | (21.9) | | |
| | Employed | 450 | (71.8) | | |
| | Retired | 40 | (6.4) | | |
| Years of education (range: 0 to 16) | | | | 9.47 | (4.00) |
| Personal monthly income (in Ghana Cedis) (range: 400 to 4500) | | | | 1938.28 | (904.46) |
| Smoking (n, %) | | | | | |
| | No | 611 | (97.4) | | |
| | Yes | 16 | (2.6) | | |
| Alcohol intake (n, %) | | | | | |
| | No | 580 | (92.5) | | |
| | Yes | 47 | (7.5) | | |
| Physical activity (range: 2.18 to 2566.20) | | | | 330.79 | (420.83) |
| Social networks (range: 0 to 30) | | | | 8.32 | (6.44) |
| Chronic physical conditions (range: 0 to 6) | | | | 1.07 | (1.35) |

N—frequency; SD—standard deviation

Chronic conditions were positively influenced by poor HRQoL (β = .352, 95%CI = .284-.421) and functional limitations (β = .272, 95%CI = .112-.433). Functional limitations were also significantly associated with increases in poor HRQoL (β = 1.213, 95%CI = 1.053–1.372). The direct effect of sleep problems on chronic conditions remained significant after including the mediators. This indicates that poor HRQoL and functional limitations partially mediated the association between sleep problems and chronic conditions.

## Bootstrapping estimates

Table 3 provides details of the direct, indirect, and total effects of the link between sleep problems and chronic conditions among single parents. The bootstrap-derived 95% CI estimation

**Table 2. Means, standard deviations, and zero-order Pearson's correlations between principal study variables.**

| | Variable | Mean | (SD) | 1 | 2 | 3 | 4 |
|---|---|---|---|---|---|---|---|
| 1 | Functional limitation | 0.15 | (0.62) | 1 | | | |
| 2 | Chronic conditions | 1.07 | (1.35) | .457*** | 1 | | |
| 3 | HRQoL | 7.33 | (1.69) | .576*** | .565*** | 1 | |
| 4 | Sleep problems | 3.14 | (0.66) | .273*** | .371*** | .436*** | 1 |

SD—standard deviation

***$p < .001$.

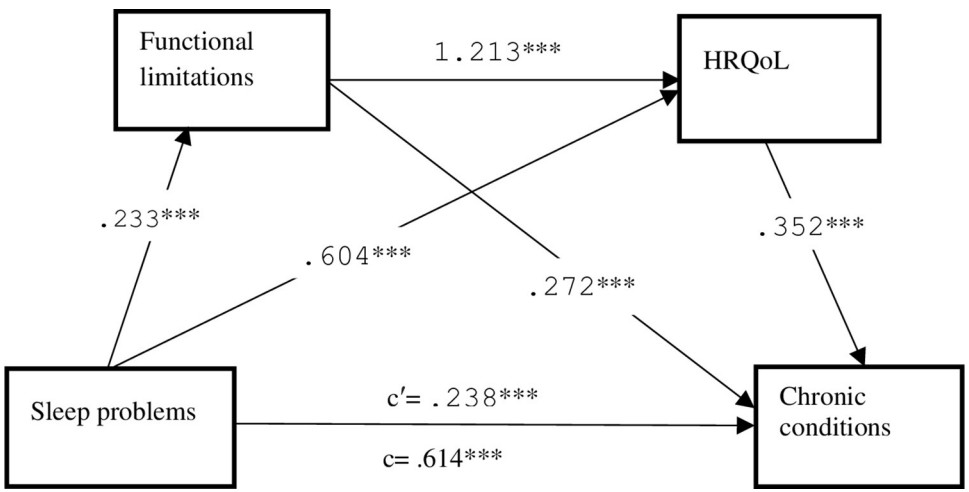

**Fig 2. A partial mediation model of the association between sleep problems and chronic conditions through HRQoL and functional limitations among single parents.** Path unstandardized coefficients are reported. Each model was adjusted for sex, physical activity, smoking status, residential type, alcohol intake, depression, NHIS status, income, religion, and education status. **$p < .005$, ***$p < .001$.

procedure with 5000 bootstrap samples did not cross zero for any estimate. Thus, a significant mediating effect of sleep problems on chronic conditions through poor HRQoL and functional limitations has been observed after controlling for several covariates. The three mediation pathways include 1) poor HRQoL ($\beta = .213$, Boot$SE = .039$, 95%CI = .143-.295) representing 34.70% of the total effect, 2) functional limitations ($\beta = .063$, Boot$SE = .029$, 95%CI = .013-.130) suggesting 10.31% of the total effect and 3) functional limitations → poor HRQoL ($\beta = .099$, Boot$SE = .025$, 95%CI = .054-.152) also suggesting 16.15% of the total effect. The total indirect or mediating effect of sleep problems on chronic conditions among single parents in this study was 61.16%.

## Discussion

### Main findings

The current study investigated the direct and indirect associations between sleep problems and chronic conditions among Ghanaian single parents in the sub-Saharan African context,

**Table 3. The indirect effects of sleep problems on chronic conditions through functional limitations and HRQoL as specific and chain mediators.**

| Path model | B | Boots*SE* | Boots 95%CI | Relative % mediated |
|---|---|---|---|---|
| Direct effect | .2384 | .0706 | .0997-.3770 | 38.84 |
| *Indirect effects* | | | | |
| SP → FL → CC | .0633 | .0299 | .0131-.1299 | 10.31 |
| SP → HRQoL → CC | .2127 | .0390 | .1433-.2954 | 34.70 |
| SP→FL→HRQoL→CC | .0994 | .0249 | .0542-.1517 | 16.15 |
| Total indirect effect | .3754 | .0595 | .2649-.4993 | 61.16 |
| Total effect | .6138 | .0753 | .4659-.7617 | 100 |

*Note*: *β*–Unstandardized regression coefficients are reported; *BootsSE*–Bootstrapping standard error; CC–Chronic conditions; SP–Sleep problems; HRQoL—Health-related quality of life; FL—Functional limitations.

Each model was adjusted for sex, physical activity, smoking status, residential type, alcohol intake, depression, NHIS status, income, religion, and education status.

The empirical 95% confidence interval does not overlap with zero.

where research on this topic is limited. The analysis found that sleep problems were independently associated with chronic conditions in a dose-response manner (showing the magnitude of the effect of sleep problems on chronic conditions) (β = .238, 95% CI = .100-.377).

Another novel finding was that functional limitations and poor HRQoL partially and serially mediate this association. To our knowledge, this is the first study to evaluate the link between sleep problems and chronic conditions among large, representative single parents in LMICs and the first to report the serial mediation effect of functional limitations and poor HRQoL in this association. Procuring policy and clinical interventions to address sleep problems may improve the physiological health outcomes of single parents, particularly in the LMICs context.

## Interpretation of findings

The findings of the current study provide valuable insights into the association between sleep problems and chronic conditions among single parents in Ghana. The analysis revealed a significant and positive association, indicating that single parents experiencing sleep disturbances are at increased risk of developing chronic health issues. The observed notable and positive correlation between sleep difficulties and chronic ailments corroborates existing research indicating the adverse effects of inadequate sleep on physical health outcomes [4]. For instance, Falkingham et al. [12] found, among 12,804 UK adults, a significant link between sleep problems and a heightened risk of chronic conditions. A review by Li et al. [3] revealed that disturbed sleep could exacerbate chronic pain, while chronic pain could disrupt sleep. Another review by Antza et al. [5] established that sleep deprivation is linked to alterations in energy regulation, insulin sensitivity, and β-cell function, thereby predisposing to obesity and type 2 diabetes mellitus (T2DM). Conversely, an extended duration of sleep is associated with chronic ailments. For instance, Nutakor et al. [4] found long sleep duration to be associated with chronic physical conditions.

A number of mechanistic pathways may explain the sleep problems-chronic conditions association. For instance, in our analysis, functional limitations explained 10.31% of the total effect of sleep problems on chronic conditions. Sleep problems, such as insomnia and sleep fragmentation, can lead to daytime fatigue, decreased energy levels, and impaired cognitive function, all of which may hinder the performance of tasks related to work, childcare, and household responsibilities [19, 20]. Moreover, the stressors associated with single parenthood, such as financial strain and social isolation, may exacerbate the adverse effects of sleep problems on functional limitations [38].

Several studies have investigated and found a positive correlation between sleep problems and functional impairment. For instance, Xiao et al. [38] showed that older individuals with poor sleep quality also exhibit psychological distress. Additionally, Vincent et al. [39] noted that older Americans may face functional challenges if they do not achieve the recommended amount of sleep. A comprehensive review and meta-analysis conducted by Amiri and Behnezhad [17] concluded that insomnia is associated with functional limitations, suggesting that improved sleep therapy could potentially enhance functional health.

The current study also provides compelling evidence of a significant and positive association between functional limitations and chronic conditions among single parents in Ghana. Functional limitations, such as mobility impairments, pain, and difficulty performing activities of daily living, can significantly impact the ability of individuals to manage chronic conditions and engage in meaningful activities [21, 22]. A potential mechanism underlying the association between functional limitations and chronic conditions among single parents is the biopsychosocial model of health. Chronic illnesses not only exert direct physiological effects but also

impact functional capacity and psychological well-being, leading to increased disability and impaired quality of life [40, 41]. Several studies support these findings. For example, Fisher et al. [18] discovered elevated odds of functional limitations among Canadians with multiple chronic conditions. Sokas et al. [41] proposed that the presence of any chronic illness in older adults correlates with the emergence of new functional limitations and poorer HRQoL post-injury. Similarly, Kim et al. [21] demonstrated that cognitive function, particularly memory and executive function, is poorer among community-dwelling Korean adults with hypertension or diabetes mellitus (DM) who have chronic illnesses compared to those without such conditions.

In this investigation, it was found that approximately 34.70% of the overall impact of sleep problems on chronic conditions could be accounted for by poor HRQoL. According to recent literature, sleep disturbances often lead to daytime fatigue, impaired cognitive function, and reduced energy levels, all of which negatively impact HRQoL [23, 24]. Indeed, people who have poor HRQoL are more likely to be battling with a number of chronic conditions [25, 26]. In a bi-directional relationship, the burden of managing chronic illnesses, coupled with the stressors associated with single parenthood, may contribute to lower HRQoL among single parents [9, 10, 15, 42].

Another novel finding of this study is that the association between sleep problems and chronic conditions was serially mediated by functional limitations → poor HRQoL by 16.15%. Literature shows that a decline in ADL can negatively impact different dimensions of HRQoL [43]. For instance, Shanbehzadeh et al. [44] demonstrated that challenges in activities of daily living were predictive of lower HRQoL among older adults aged 60 to 90 years following coronavirus infection. Additionally, Lee et al. [45] observed that functional status significantly influenced poor HRQoL among Australian patients with multiple chronic conditions.

## Implications of the study

The study's findings underscore the importance of incorporating sleep health initiatives into public health policies targeted at single-parent households in LMICs. Policies should prioritize the development and implementation of comprehensive sleep health programs tailored to the unique needs and challenges faced by single parents. These initiatives may include community-based education campaigns on sleep hygiene practices, access to affordable and culturally sensitive healthcare services for sleep disorders, and workplace policies that support work-life balance and adequate rest for single parents. Moreover, efforts to improve functional capacity and HRQoL among single parents may serve as potential avenues for mitigating the adverse health effects of sleep problems and reducing the burden of chronic diseases in this population.

The study's findings highlight the importance of healthcare providers integrating sleep assessments into routine healthcare evaluations for single parents, particularly in LMICs where healthcare resources may be limited. Screening tools for sleep disorders should be incorporated into primary care settings to facilitate early detection and intervention. Based on the novel finding of the serial mediation by functional limitations and poor HRQoL, interventions aimed at improving HRQoL and addressing functional limitations may serve as effective strategies for mitigating the adverse health effects of sleep problems among single parents. Psychosocial support programs that address the unique stressors associated with single parenthood and promote coping mechanisms for managing sleep disturbances could be particularly beneficial. As the first study to evaluate the link between sleep problems and chronic conditions among a large, representative sample of single parents in LMICs, this research highlights the need for further investigation to explore potential moderators and longitudinal effects.

## Strengths and limitations

This study stands out as one of the few investigations to explore the relationship between sleep problems and chronic conditions, marking the first attempt to investigate the sequential mediation of functional limitations and poor HRQoL in this association among single parents in Ghana. Given the limited existing literature on sleep problems among single parents in Ghana, the findings from this study offer foundational data for future research endeavors. Our primary variables, informed by prior research, were assessed using established scales with robust reliability and content validity. Additionally, several potential confounding factors were accounted for to enhance the validity of the findings. Despite these strengths, the study had some limitations. Limited sample size especially the selected communities for a study of such a scale. Again, the utilization of a cross-sectional design impedes the ability to establish causal relationships. Furthermore, reliance on retrospective self-reporting through the research instrument (questionnaire) may introduce recall and reporting biases. Future studies should consider objectively evaluating sleep problems, chronic conditions, functional limitations, and poor HRQoL to bolster the validity of the results. Nonetheless, self-reporting remains a valuable method for capturing respondents' subjective assessments of their physiological and psychosocial status.

## Conclusions

This study examined the extent to which functional limitations and poor HRQoL serially explain the sleep problems and chronic conditions linkage among single parents in Ghana. Sleep problems were positively associated with chronic conditions, suggesting that single parents going through sleep problems are likely to suffer chronic conditions. The results further suggest that poor HRQoL and functional limitations significantly mediated the association between sleep problems and chronic conditions. Strategies to increase sleep quality among single parents may potentially reduce poor HRQoL and functional limitations. Future studies may consider longitudinal design for the possibility of drawing causal conclusions.

## Supporting information

**S1 Dataset. Supporting information of dataset including the study variables.**
(SAV)

## Acknowledgments

The authors thank Shadrach Owusu, Joyceline Owusu, and Matilda Owusu for their assistance with data collection.

## Author Contributions

**Conceptualization:** Obed Jones Owusu-Sarpong, Razak M. Gyasi.

**Data curation:** Obed Jones Owusu-Sarpong, Kabila Abass, Daniel Buor, Solomon Osei Tutu, Razak M. Gyasi.

**Formal analysis:** Obed Jones Owusu-Sarpong, Razak M. Gyasi.

**Supervision:** Razak M. Gyasi.

**Writing – original draft:** Obed Jones Owusu-Sarpong, Kabila Abass, Daniel Buor, Solomon Osei Tutu, Razak M. Gyasi.

**Writing – review & editing:** Obed Jones Owusu-Sarpong, Kabila Abass, Daniel Buor, Solomon Osei Tutu, Razak M. Gyasi.

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
