## [Decision Letter · Decision Letter 0]

20 Sep 2024

PONE-D-24-19385Single parenthood, sleep, and chronic conditions in Ghana: The importance of health-related quality of life and functional limitationsPLOS ONE

Dear Dr. Owusu-Sarpong,

Thank you for submitting your manuscript to PLOS ONE. After careful consideration, we feel that it has merit but does not fully meet PLOS ONE’s publication criteria as it currently stands. Therefore, we invite you to submit a revised version of the manuscript that addresses the points raised during the review process.

**ACADEMIC EDITOR: Dear Author, please attned to all the comments provided by the reviewer/s.**

We look forward to receiving your revised manuscript.

Kind regards,

Zulkarnain Jaafar

Academic Editor

PLOS ONE

2. Please provide additional details regarding participant consent. Your study included minors, state whether you obtained consent from parents or guardians. If the need for consent was waived by the ethics committee, please include this information.

3. In this instance it seems there may be acceptable restrictions in place that prevent the public sharing of your minimal data. However, in line with our goal of ensuring long-term data availability to all interested researchers, PLOS’ Data Policy states that authors cannot be the sole named individuals responsible for ensuring data access (http://journals.plos.org/plosone/s/data-availability#loc-acceptable-data-sharing-methods).

Reviewers' comments:

Reviewer's Responses to Questions

**Comments to the Author**

1. Is the manuscript technically sound, and do the data support the conclusions?

Reviewer #1: Yes

Reviewer #2: Yes

2. Has the statistical analysis been performed appropriately and rigorously? 

Reviewer #1: Yes

Reviewer #2: I Don't Know

3. Have the authors made all data underlying the findings in their manuscript fully available?

Reviewer #1: Yes

Reviewer #2: Yes

4. Is the manuscript presented in an intelligible fashion and written in standard English?

Reviewer #1: Yes

Reviewer #2: Yes

5. Review Comments to the Author

Reviewer #1: This study is relevant and was presented in an intelligible fashion. The introduction was engaging and sets a strong foundation for the article. The conclusion and implications also tie back to the main points and offers insights for future research.

Reviewer #2: The findings of this study can inform public health policies to improve sleep hygiene in countries where single parenthood is prevalent. I suggest some comments for improved readability of the paper.

Title

The title seems to suggest that single parenthood is the variable of interest in this study, which is not the case. In fact, single parents are the study population. Please change the title to reflect the study design.

Abstract

Please do not introduce an acronym without spelling the full term at its first appearance.

“no study has reported the serial mediation effects of physiological factors in this association.” I am not sure if I would categorize HRQoL and functional limitations as physiological factors.

“the hypothesized association”? Where are the hypotheses?

Methods

Is the dependent variable the number of chronic conditions? Please explain clearly.

Why did the authors use the sum of EQ-5D scores instead of the EQ-5D index, which can be calculated using the value set and is commonly used as a continuous variable?

I am not sure if you can sum EQ-5D responses (1, 2, to 3) and use the sum as a continues variable to conduct Pearsons correlation and OLS. That is because EQ-5D responses are not an interval scale variable.

Lines 138-139. If 6 items are measured on 1 or 2, the total score should range from 6 to 12. Please check the accuracy of the statement.

Discussion

This section is a bit wordy. For example, can you summarize the review of the literature (Line 243-262), which seems a bit lengthy?. In general, streamlining the discussion would enhance readability.

Specific comments

Line 83, administered is duplicative

Line 108, what does the acronym SSA stand for?

Line 153 The first mention of the acronym should be explained.

Line 166, Cis � Cis

Line 173, confidence intervals � Cis

Line 182, “gainfully employed” sounds like a colloquial expression. Can you just use “employed”?

Line 185, can you add the unit for physical activity?

Line 231, What do you mean by “in a dose-response manner”?

Line 324, check the reference style.

6. PLOS authors have the option to publish the peer review history of their article (what does this mean?). If published, this will include your full peer review and any attached files.

Reviewer #1: **Yes: **Araba Aseye Ahiabu

Reviewer #2: No

---

## [Author Response · Author response to Decision Letter 0]

27 Sep 2024

Reviewer comments 

Reviewer #1: 

This study is relevant and was presented in an intelligible fashion. The introduction was engaging, and set a strong foundation for the article. The conclusion and implications also tie back to the main points and offer insights for future research.

Response: We thank the Reviewer for the compliment. We take note of this positive comment with much gratitude, particularly regarding the intelligible nature of our study.

Reviewer #2: 

The findings of this study can inform public health policies to improve sleep hygiene in countries where single parenthood is prevalent. I suggest some comments for improved readability of the paper. 

Response: We appreciate the observation that our study can inform public health policies to improve sleep hygiene among this vulnerable population. We would like to thank the reviewers for a nice summary of our study and for revealing the potential interest of our paper in contributing to the academic and policymaking communities. We have taken note of all the constructive comments provided and have attempted to improve the quality of the paper further, guided by these critical comments..

Title

The title seems to suggest that single parenthood is the variable of interest in this study, which is not the case. In fact, single parents are the study population. Please change the title to reflect the study design.

Response: Thank you for this crucial observation. The title has been revised as follows;

 “Sleep problems and chronic conditions in single parents in Ghana: Serial mediating roles of health-related quality of life and functional limitations.”

Abstract

Please do not introduce an acronym without spelling the full term at its first appearance.

Response: Acronyms such as low—and middle-income countries (LMICs) and health-related quality of life (HRQoL) have been spelled out fully in their first use. Please see Page 2 of the abstract.

“No study has reported the serial mediation effects of physiological factors in this association.” I am not sure if I would categorize HRQoL and functional limitations as physiological factors.

Response: Thank you for this. Physiological factors have been replaced with functional limitations and poor health-related quality of life. Please see page 2 of the abstract.

“The hypothesized association”? Where are the hypotheses?

Response: We have stated two hypotheses in the introduction section of the revised paper (see page 5). We have indicated that: 

1) We hypothesized that there would be significant positive association between sleep problems and chronic conditions 

2) We further hypothesized that the positive association between sleep problems and chronic conditions would be serially mediated by functional limitations and poor health-related quality of life.

Methods

Is the dependent variable the number of chronic conditions? Please explain clearly.

Response: The dependent variable is chronic conditions, which consist of 11 chronic physical conditions. Please see page 8.

Why did the authors use the sum of EQ-5D scores instead of the EQ-5D index, which can be calculated using the value set and is commonly used as a continuous variable?

I am not sure if you can sum EQ-5D responses (1, 2, to 3) and use the sum as a continuous variable to conduct Pearson's correlation and OLS. That is because EQ-5D responses are not interval scale variables.

Response: We truly understand the reviewer's drift in this regard. We would like to humbly say that, like the EQ-5D-5L, the EQ-5D-3L has been standardized and emanated a continuous index that has been used to perform linear regressions and Pearson’s correlations in many previous studies. The credibility of such an index has been demonstrated. 

Lines 138-139. If 6 items are measured on 1 or 2, the total score should range from 6 to 12. Please check the accuracy of the statement.

Response: Thank you for this observation. We scored the responses of each of the six items on a binary (no=1 or yes=2) scale. This means that the total minimum score will range from 6 to 12, as you rightly indicated. We have revised and included this in the revised manuscript. Please see page 9. 

Discussion

This section is a bit wordy. For example, can you summarize the review of the literature (Line 243-262), which seems a bit lengthy? In general, streamlining the discussion would enhance readability.

Response: We have summarized the literature review in the discussion section. Thank you very much. The summary is done on page 14.

Specific comments

Line 83, administered, is duplicative

Response: This has been rectified on page 6. 

Line 108, what does the acronym SSA stand for?

Response: The acronym SSA stands for sub-Saharan Africa. 

Line 153, the first mention of the acronym should be explained.

Response: The acronyms NHIS and SPSS stand for National Health Insurance Scheme and Statistical Package for Social Scientists, respectively. We have defined these fully in their first use.

Line 166, Cis � Cis

Response: It now reads confidence intervals (CIs) on page 10

Line 173, confidence intervals � Cis

 Response: It is now consistent as confidence intervals (CI). Please see page 10.

Line 182, “gainfully employed,” sounds like a colloquial expression. Can you just use “employed”?

Response: Thank you for this correction. This has been done on page 10.

Line 185, can you add the unit for physical activity?

Response: The unit for physical activity has been added as 330.79 MET min

Line 231, what do you mean by “in a dose-response manner”?

Response: “a dose-response manner” has been explained as showing the magnitude of sleep problems' effect on chronic conditions. Please see page 13.

Line 324, check the reference style.

Response: It has been done on page 16.

We are grateful for your time spent reviewing and offering essential suggestions and additions that have significantly improved our paper. We also thank you for giving us another opportunity to revise and resubmit it to Plos One. 

Yours sincerely,

(Corresponding author)

---

## [Decision Letter · Decision Letter 1]

4 Oct 2024

Sleep problems and chronic conditions in single parents in Ghana: Serial mediating roles of health-related quality of life and functional limitations

PONE-D-24-19385R1

Dear Dr.  Owusu-Sarpong,

We’re pleased to inform you that your manuscript has been judged scientifically suitable for publication and will be formally accepted for publication once it meets all outstanding technical requirements.

Kind regards,

Zulkarnain Jaafar

Academic Editor

PLOS ONE

Additional Editor Comments (optional):

Reviewers' comments:

Reviewer's Responses to Questions

**Comments to the Author**

1. If the authors have adequately addressed your comments raised in a previous round of review and you feel that this manuscript is now acceptable for publication, you may indicate that here to bypass the “Comments to the Author” section, enter your conflict of interest statement in the “Confidential to Editor” section, and submit your "Accept" recommendation.

Reviewer #2: All comments have been addressed

2. Is the manuscript technically sound, and do the data support the conclusions?

Reviewer #2: Yes

3. Has the statistical analysis been performed appropriately and rigorously? 

Reviewer #2: Yes

4. Have the authors made all data underlying the findings in their manuscript fully available?

Reviewer #2: (No Response)

5. Is the manuscript presented in an intelligible fashion and written in standard English?

Reviewer #2: Yes

6. Review Comments to the Author

Reviewer #2: (No Response)

7. PLOS authors have the option to publish the peer review history of their article (what does this mean?). If published, this will include your full peer review and any attached files.

Reviewer #2: No

---

## [Editor Report · Acceptance letter]

18 Oct 2024

PONE-D-24-19385R1 

PLOS ONE

Dear Dr. Owusu-Sarpong, 

I'm pleased to inform you that your manuscript has been deemed suitable for publication in PLOS ONE. Congratulations! Your manuscript is now being handed over to our production team.

Kind regards, 

on behalf of

Dr. Zulkarnain Jaafar 

Academic Editor

PLOS ONE